# Assessing the Foodshed and Food Self-Sufficiency of the Pearl River Delta Megacity Region in China

**DOI:** 10.3390/foods12234210

**Published:** 2023-11-22

**Authors:** Yankai Wang, Haochen Shi, Yuyang Zhang, Xinjian Li, Miaoxi Zhao, Binbin Sun

**Affiliations:** 1School of Architecture, State Key Laboratory of Subtropical Building and Urban Science, South China University of Technology, Guangzhou 510641, China; arwayork666@mail.scut.edu.cn (Y.W.); ar_shc@mail.scut.edu.cn (H.S.); 202220104946@mail.scut.edu.cn (Y.Z.); lxj20229093@scut.edu.cn (X.L.); 2School of Social Sciences, University Sains Malaysia, Gelugor 11800, Malaysia; sunbinbin@student.usm.my

**Keywords:** megacity region, food self-sufficiency, foodshed, food security, food system resilience, Pearl River Delta

## Abstract

Food self-sufficiency has long been regarded as essential for understanding and managing urban and regional food systems; however, few studies have examined the food self-sufficiency of megacity regions within a comprehensive framework that distinguishes different types of agricultural land (i.e., arable land, horticultural landscapes, and waters). To fill these gaps, we took the Pearl River Delta as a case study and quantified the foodsheds of different types of agricultural land by calculating the land footprint of food consumption. On this basis, food self-sufficiency is defined as the ratio of available and required agricultural area for regional food demand. The results indicated that the self-sufficiency level provided by the arable land in the Pearl River Delta is low and cannot realize self-sufficiency at the regional and urban levels. The horticultural landscapes can provide self-sufficiency at the regional level, whereas the regions with water cannot, as their foodsheds extend over the boundary of the Pearl River Delta. For arable land, establishing a localized regional food system requires expanding the foodshed size. These findings provide evidence that megacity regions may face increasing difficulties in achieving self-sufficiency in the near future. This research can improve policymakers’ understanding of the sustainability and resilience of regional food systems in megacity regions.

## 1. Introduction

The food self-sufficiency of megacities has gradually become a key topic of concern in the past few decades [1,2,3]. Currently, more than half of the global population lives in urban areas, and this proportion will continuously increase to two-thirds in 2050, according to a forecast from the United Nations [4]. Rapid population growth that is accompanied by the conversion of agricultural land into built-up urban areas has led to a significant and concerning shortage of available agricultural land, which has become a critical issue for urban food security [5,6]. Facing the challenges of ongoing global urbanization and metropolitan areas growth, the discussions concerning regional and localized food systems are continuously increasing, with the widely acknowledged fact that cities indeed rely on their adjacent rural hinterlands for the supply of food [7,8,9]. In this context, food self-sufficiency, defined as the capacity of cities/regions to obtain sufficient food within their boundaries [10,11,12], has received increasing attention due to its heightened importance in enhancing the sustainability and resilience of localized food systems [11,13,14]. As has been demonstrated by previous studies, increasing food self-sufficiency contributes greatly to shortening food supply chains and decreasing greenhouse gas emissions, as well as providing an opportunity for implementing sustainable management practices in agriculture (e.g., organic agriculture), thereby enhancing the resilience and sustainability of food systems [12,15,16].

The foodshed concept has gradually gained attention due to its importance as a tool for understanding urban and regional self-sufficiency [9,12,17]. This concept can aid in determining the area of agricultural land that a city or region would theoretically need to meet self-sufficiency [7,18]. In the early 20th century, MacKaye [19] studied the food supply in Washington D.C. to explore logistics efficiency. To our knowledge, this is also the first empirical study of the linkages between cities and agricultural hinterlands through the supply chain. Following their idea, some scholars proposed a new concept—the foodshed—to understand the food supply situation of a targeted area [13,18]. These scholars also regarded this concept as helping to establish a sustainable and desirable local food system [18,20]. In recent years, an extensive body of literature studies has highlighted the concept’s utility as a quantitative framework for analyzing urban food supply and urban–rural linkages [20,21,22]. Vicente-Vicente et al. [22], for instance, have pointed out that a foodshed can be considered as the territory around an urban/metropolitan area that is required to feed its population. In other words, foodsheds indicate the connections between an urban/metropolitan area and its surrounding agricultural production hinterland [23,24]. Zasada et al. [9] proposed the Metropolitan Foodshed and Self-Sufficiency Scenario (MFSS) model with an assessment capacity at the city level. Following their work, numerous studies extended the MFSS and similar tools to the regional level to support specific strategies for managing localized food systems [12,25,26]. Such approaches to defining and implementing the foodshed concept can help us better understand the underlying factors that contribute to urban/regional self-sufficiency in the context of urbanization [26]. In particular, because a foodshed naturally has spatial characteristics, it can provide information about the possibility of satisfying local demand through adjacent agriculture, especially when the foodshed boundaries are beyond the borders of the targeted analyzed region [9,13]. All in all, foodshed research can provide a tool with which to assess and understand relationships between local food self-sufficiency and land-related factors (e.g., fertilizer use and land use). Thereby, it can be said that foodsheds can contribute to food systems’ sustainability, as well as to identifying and mapping food flows (e.g., food imports or food exports) that result in interregional interdependencies and can influence the supply chain resilience [20]. Hence, it is necessary to map foodsheds and utilize such information to evaluate food self-sufficiency as well as the possibility of localized targeted food production.

Although the importance of foodsheds and food self-sufficiency are well acknowledged, the assessment and characterization of their constituents still have two major shortcomings and are ripe for further improvements and explorations. The first pertains to the lack of input on the different land types during the foodshed calculation. Many quantitative assessments have been conducted based on food production and consumption to understand regional self-sufficiency and foodsheds [9,27,28]. These studies usually calculate the total amount of food consumption as well as production and subsequently convert them into the required as well as the available agricultural land resources [9,12,21]; however, such assessments of agricultural land that do not consider the weighting of different land types are not sufficient to comprehensively evaluate the self-sufficiency levels of urban areas, as there is a possibility that a city or a region could be self-sufficient when considering a specific type of agricultural land but not another. In this context, when evaluating food self-sufficiency and foodsheds, the categorization of different types of agricultural land is needed (i.e., arable land, horticultural landscapes, and waters).

The second shortcoming concerns the research objects, or, in other words, the areas of study. The majority of existing studies have assessed self-sufficiency and foodsheds at different scales, including national, regional, or urban scales [9,12,25,29]; however, to the best of our knowledge, no known studies have assessed megacity regions with high population concentrations and multiple competing central cities [30,31,32]. In recent years, as globalization accelerates, megacity regions are continuously springing up across the world and are labeled as the future outcome of the urbanization process and the primary direction of urban expansion [33,34]. Typical examples of megacity regions (MCRs) are the Pearl River Delta and Yangtze River Delta regions in China, the Tokaido (Tokyo–Osaka) corridor in Japan, and Greater Jakarta in Indonesia [34,35]. Liu et al. [36] have pointed out that there is a huge gap between local food supply and demand due to the large populations and limited agricultural land in megacity regions. This implies that the food supply of MCRs is more dependent on their efficient connections with their hinterland, and this results in significant challenges for achieving food self-sufficiency [9,12,37]. Additionally, due to continuous population growth, especially in MCRs where multiple central cities exist, the foodshed area of each is likely to overlap [7]. This causes competition for the same agricultural land resources and hence creates additional challenges to regional self-sufficiency. In this context, MCRs can provide a more comprehensive perspective for understanding the complexity of foodshed and urban–regional food systems.

To fill the two abovementioned knowledge gaps, we conducted a comprehensive food self-sufficiency and foodshed assessment in a typical MCR, the Pearl River Delta (PRD), including an in-depth consideration of key factors that have not been adequately studied, such as the distinguishment of different agricultural land types. The food consumption of the PRD was calculated through the lens of different agricultural land types, and the corresponding foodshed and self-sufficiency characteristics were analyzed. The scientific contributions of this study can be summarized in three key findings: Firstly, we developed a new framework for exploring regional self-sufficiency by distinguishing different agricultural land types and provided new substantive insights for the comprehensive characterization of regional self-sufficiency. Secondly, a practical case study was conducted in the PRD, a typical MCR, to provide evidence on whether self-sufficiency can be achieved in the MCR when different types of agricultural land are considered. Finally, mapping the foodshed revealed details of localized food production capacity with regard to the different land resources in the MCR, providing the capacity to build up spatially defined local food systems in conjunction with the adjacent agricultural capacity. To conclude, based on a comprehensive understanding of the MCR’s self-sufficiency and foodshed, we can more comprehensively understand the sustainability and resiliency of local food systems and, therefore, put forward policies for improving their capacity and efficiency.

The remainder of the paper is organized as follows: Section 2 introduces the study area, data, and methods. In Section 3, we calculate the food consumption of the PRD through the lens of different types of agricultural land and analyze the corresponding foodshed as well as self-sufficiency characteristics. In Section 4, we discuss the case study’s findings, the relevant improved policies, research limitations, and future research directions. Finally, the study’s conclusions are provided in Section 5.

## 2. Materials and Methods

### 2.1. Study Area and Data

The Pearl River Delta megacity region (PRD-MCR), located in the south of China, is one of China’s most populous and economically developed regions. The PRD-MCR comprises nine prefectural-level cities: Guangzhou, Shenzhen, Foshan, Dongguan, Zhuhai, Zhongshan, Huizhou, Zhaoqing, and Jiangmen (Figure 1). PRD has a total population of over 78 million people, according to the 2020 consensus, and a total area of 55,368.7 km^2^, of which hills, mountains, and islands account for about 30%. Due to accelerated urbanization during the past half-century, under the trend of globalization, the PRD has become an MCR, in which the population and its density have dramatically increased, and large areas of agricultural land have been converted into built-up urban areas [32,38]. The PRD-MCR, as it has a high population density and less agricultural land, faces the most difficult challenges with regard to regional food system sustainability compared to those of other regions [39,40].

The data used in this study are (1) food consumption data, (2) population data, and (3) land cover data. Detailed information on the data is shown in Table 1.

### 2.2. Methodology

Generally, foodshed assessments are accomplished by defining agricultural capacity, food flows, or the combination of both (hereinafter referred to as hybrid) [20,22,25]. Specifically, agricultural capacity is an assessment of the agricultural production capability that ensures a local population’s food demands [21]. Firstly, capacity studies, which juxtapose local food production and consumption, are the most common methods used to estimate food self-sufficiency potential and the size of a foodshed to meet local food demands [20]. The second approach tracks the information on food flows between food production and consumption areas [43]. Finally, hybrid studies are carried out to calculate the food production and consumption ratio while accounting for food flows (food imports and exports) on different spatial scales, enabling an analysis of interdependencies between regions [20]. Considering the difficulties in data acquisition, many researchers tend to employ capacity studies to define foodsheds [9,12].

According to the review above, we assess the self-sufficiency and size of a foodshed in PRD-MCR based on capacity approaches. In capacity assessments, the theoretical food–land footprint and the potential self-sufficiency are evaluated by considering the population and current dietary patterns (food consumption), the farmland available, land-use cover, and regional yields [12]; therefore, we first calculated the hypothetical agricultural land requirements for each type of food (food–land footprint) to determine the foodshed (Figure 2). The self-sufficiency level was then assessed by comparing the foodshed area with the utilizable agricultural area within the boundary of a predefined geo-object. The detailed assessment of the foodshed and food self-sufficiency was divided into four steps.

The first step was to identify the major agricultural land types and their food production capacity based on different food consumption scenarios. We regarded arable land, horticultural landscapes, and waters as the major three types of agricultural land used for the foodshed analysis, as these land types essentially cover all of the necessary food production. It should be noted that another significant food-producing land type—grassland—was excluded from our assessment, as the livestock sector in the MCR is not based on grassland for animal feeding but rather on feed produced from arable land.

The second step was to establish rough quantitative connections between food consumption and its corresponding land area in the three agricultural land types that we identified previously. To achieve this, we initially identified the types of agricultural land accounting for the production of specific food products or food sectors. Then, based on this relationship, we estimated the unit of agricultural land area needed per kg of food produced, using as a reference general crop yields and feed conversion ratios for meat, eggs, and milk [44,45]. Finally, by integrating this information we acquired quantitative connections bridging the food consumption per capita and the area demand of agricultural land per capita (Table 2). Accordingly, the specific calculation of food, *i* or *j*, for this process was as follows:(1)$$LD_{i}=\left\{\begin{array}{c} FC_{i}\times \omega_{i}\\ FC_{i}\times \frac{1}{\varphi_{i}}\times \omega_{j} \end{array}\right.$$


*i* corresponds to cereals, oil crops, roots and tubers, sugars, legumes, fruits, or vegetables.*j* corresponds to animal products (e.g., beef, mutton, pork, poultry, dairy, or eggs) and aquatic products.


In Equation (1), *LD_i_* represents the agricultural land area demand per capita; *FC_i_* represents food consumption per capita (kg); and *ω* is the conversion factor of crop production to the agricultural land, representing the agricultural land area demand per kg of food production. *ω* can be calculated based on the yield per hectare of the different crops; *φ_i_* represents the feed conversion rate, i.e., the amount of edible animal meat produced per kg of feed (excluding bones and some viscus tissues).

The third step included the calculation of the agricultural land area (*LD_total_*) that can meet the needs of the total population of the PRD (*POP_total_*):(2)$${LD}_{total}={LD}_{i}{\times POP}_{total}$$
where the population number is derived from the China City Statistical Yearbook [42].

The final step was to define the regional foodshed and its food self-sufficiency. A foodshed of a geo-object (e.g., a city) can be represented as a circle that is centered on the centroid of its administration [9,12]. The size and spatial range of the circle are determined by the agricultural land area needed to meet the food consumption demands of this geo-object. More precisely, the foodshed’s detailed definition or spatial localization is based on two assumptions: The first is that the agricultural land area within a geo-object can meet the food consumption demands of its local population. In this case, only the agricultural land within the geo-object will be used to define the foodshed. On the contrary, when the agricultural land area within the geo-object cannot meet the food consumption demands of its local population, the agriculture land neighboring the geo-object should be used to meet the food demand of its local population. In this case, the size of the foodshed will be defined based on the area of the geo-object’s internal and adjacent external agricultural land.

To obtain the foodshed (circle), we developed a recursive function to calculate its size and spatial range. Considering the fact that the foodshed essentially presents as a circle on a map, a circle was used for its representation in the following steps. Specifically, we used geo-object A as an example with which to demonstrate the whole process. Firstly, we calculated the centroid of geo-object A as the center of the circle. Based on this point, we then established a small circle with a radius of 1 km and calculated the total area of agricultural land patches within this circle. Subsequently, we estimated whether the total area that we calculated was approximately equal to the required agricultural land for the food demands of geo-object A. If so, we can acquire a circle polygon to represent the foodshed. If the area could not cover the food demands, we needed to increase the circle’s radius, and the total area covering agricultural land patches was recalculated to re-estimate whether the above approximate equation was tenable. This circulation-like process was carried on and ended only when the answer to the (re)estimate was yes. To better understand the entire procedure, we utilized a pseudocode for its demonstration. The Algorithm 1 is as follows:
**Algorithm 1** Pseudocode demonstrating the foodshed determination.  **Input:** total demand of agricultural area (*A_total.dem_*),           a centroid of the administrative boundary polygon (*Centroid*),          spatial distribution of agricultural land patches  **Output:** radius of a circle (foodshed) (*rFS*) 
   **define a thresholds**  **if**   *A_total.dem_* for local population food demand can be met within the boundaries then       as the center of the circle with *Centroid*,       define *rFS* = 1 km      **sum** agricultural area patches within the circle      **while**
*A_total.dem_* − total area of agricultural land patches > thresholds                   radius = radius + 0.01 km       **end**       **output**
*rFS*
 **else**   *A_total.dem_* for local population food demand can’t be met within the boundaries then       as the center of the circle with *Centroid,*       define *rFS* = 1 km      **sum** agricultural area patches within the circle       **while** *A_total.dem_* − total area of agricultural land patches > thresholds                   radius = radius + 0.01 km       **end**       **output**
*rFS* **end**

Based on the above foodshed determination, the level of food self-sufficiency is expressed as the ratio between the total demand for agricultural land to meet food consumption and the available agricultural land within the geo-object [9]. In general, if the self-sufficiency of a geo-object is equal to or higher than 100%, it means that the food production within this geo-object can meet its population demand and can also have an underlying food export capacity. On the contrary, this geo-object will inevitably need food imports if the sufficiency is less than 100%.

## 3. Results

### 3.1. Agricultural Area Demand

The consumption of each type of food in the PRD differs significantly, resulting in differences in the corresponding agricultural land demands (Figure 3). Cereals, vegetables, and animal products are highly consumed in the PRD, accounting for approximately 85% of the overall diet. In addition, the fruit consumption proportion has also reached 11%. Regarding agricultural land demand, arable land is the major land type for food production. The total demand for arable land in the PRD is as high as 71,509.28 km^2^, accounting for 93% of the overall demand (Table 3). Similarly, the demand for horticultural land and water bodies is 1986.94 km^2^ and 3296.93 km^2^, respectively. In addition, when focusing on the city level, central cities with a high population, such as Guangzhou, Shenzhen, Dongguan, and Foshan, account for the largest share of the demand for various types of food and the associated agricultural land. In contrast, cities with a low population, such as Zhuhai and Zhaoqing, have a lower demand for both.

### 3.2. PRD-MCR Foodshed Analysis

The foodsheds for the different agricultural land types in the PRD differ significantly due to the widely varying demands on the types of agricultural land. In terms of arable land, the PRD requires approximately 71,509.28 km^2^ to achieve self-sufficiency, and the radius of its foodshed is up to 397.06 km. The foodshed extends from the PRD to areas east, west, and north of Guangdong, and even covers certain areas of neighboring provinces, such as Guangxi, Hunan, and Jiangxi (Figure 4a). Compared to the PRD, the populations in these neighboring areas are relatively small, so these areas can theoretically support the demand for arable land. Moreover, the foodshed of horticultural areas was the smallest, with a radius of 40.97 km, and was completely contained within the PRD (Figure 4b). This indicates that the horticultural areas within the PRD can satisfy the population’s food demand. Regarding the water bodies, their foodshed had a radius of 143.56 km and covered cities such as Qingyuan, Shaoguan, and Heyuan on the periphery of the PRD (Figure 4c). These areas have relatively small populations and abundant water bodies, and thus are capable of achieving self-sufficiency in the PRD.

As for the city-level foodsheds, there were significant differences regarding the agricultural land types in different cities. The larger foodsheds were in densely populated cities such as Guangzhou, Shenzhen, Dongguan, and Foshan. In terms of arable land, each city’s foodshed not only completely covers its administration area but also extends to that of its neighboring cities. The average radius of the foodsheds is approximately 115.40 km. Among them, the foodsheds in Zhaoqing and Jiangmen are the smallest in size. Shenzhen and Guangzhou significantly contribute to the food consumption of the MCR due to their large populations. At the same time, these two cities have insufficient arable land resources, and as a result both have the largest foodshed size. Notably, although Foshan does not have the highest demand for horticultural crops among the cities due to the scarcity of regions where horticultural crops are grown within its administration, more adjacent areas need to be included in its foodshed to satisfy its demand. In addition, the population of Zhuhai is the smallest among all of the cities, and, as a result, its water body foodshed was the smallest compared to that of other cities.

### 3.3. Regional Self-Sufficiency

As shown in Figure 4 and Table 3, only regions with horticultural crops can achieve theoretical self-sufficiency among the three types of agricultural land. The self-sufficiency rate of horticultural regions reached 214.96%, reaching theoretical self-sufficiency but also having the capability to export their production. On the contrary, the self-sufficiency rate of the water body foodsheds was 84.13%, not reaching self-sufficiency. Compared with regions with horticultural crops and water bodies, the self-sufficiency rate of the arable land was the lowest, at only 16.73%. This means that the available arable land in the PRD is significantly lower than that needed for self-sufficiency, and most of the population needs to rely on the importing of food to support themselves.

A city-level analysis is presented in Figure 4. The cities with the largest populations, such as Shenzhen, Dongguan, Foshan, and Guangzhou, had lower self-sufficiency rates. More specifically, Shenzhen, Dongguan, and Foshan cannot achieve self-sufficiency in all three types of agricultural land. Regarding the arable land, all of the cities cannot achieve self-sufficiency. Due to their large populations and scarce available arable land, Shenzhen and Dongguan, whose self-sufficiency rates are only 0.75% and 2.70%, respectively, face the greatest challenges in terms of food supply. All of the cities, except Guangzhou, Foshan, Shenzhen, and Dongguan, can achieve theoretical self-sufficiency regarding the water body foodsheds. Regarding regions with horticultural crops, cities such as Jiangmen, Zhaoqing, and Huizhou exhibit relatively high self-sufficiency rates. Surprisingly, Guangzhou has a 162.96% self-sufficiency rate with regard to horticultural crop production. This is because the urban area of Guangzhou is relatively large, and regions with horticultural crops in Zengcheng and Conghua, on the periphery of Guangzhou, are abundant, making these areas important producing regions of subtropical fruits such as lychees, mangoes, and bananas. In addition, due to the small area with horticultural crops in Foshan and its large population, its self-sufficiency rate is only 13.69%, which is the lowest among all of the cities.

## 4. Discussion

Given the increasing interest in regional food systems, it has become a prerequisite to explore regional foodsheds, self-sufficiency, and localized food systems [11,29]; however, existing studies have only provided partial information on different spatial scales at the national, regional, and urban levels [9,12,21,46]. At the same time, not enough attention has been given to megacity regions and a greater emphasis on the different agricultural land types, potentially leading to a lack of a comprehensive understanding of regional foodsheds and self-sufficiency. Therefore, focusing on the PRD-MCR, we evaluated its self-sufficiency levels and the foodshed capacity of different agricultural land types.

Firstly, the results of this study present a comprehensive analysis of regional self-sufficiency. Compared to the evaluation of all agricultural land types as a whole, significant differences were observed in the self-sufficiency levels of the three different agricultural land types in the PRD; however, self-sufficiency cannot be achieved from arable land and water body production, while it is achieved in the horticultural regions, which, due to the surplus, also have the capacity for food exports. Our research results share similarities with the findings of Li et al. [40] in analyzing food self-sufficiency in eastern China—that is, one region could be self-sufficient in one food commodity but not another. Along with their findings, we point out the importance of distinguishing between different types of agricultural land for estimating self-sufficiency and the substantive insights that it provides for self-sufficiency estimation at the regional level.

Secondly, the empirical results from the PRD region indicate the growing difficulty in achieving self-sufficiency in the MCR. Only regions with horticultural crops could provide a theoretical self-sufficiency among the three types of agricultural land. The self-sufficiency rate of the water bodies was 84.13%, while that of arable land was only 16.73%. These results demonstrate the considerably serious supply and demand gaps in terms of arable land and arable crop production in the PRD. In the forthcoming years, it is projected that the population of MCRs across the world will continue to increase [33,47]. This further implies that the trend of a decreasing self-sufficiency rate regarding various types of agricultural land and their production may become more pronounced, and that the MCR may face increasing difficulties in achieving self-sufficiency. In addition, our results are in line with the findings of Sylla et al. [21], who point out, in their investigations of nine metropolitan areas in European countries, that more than half of them cannot achieve self-sufficiency. Although there are differences in the research objective, the trend of increased difficulty in achieving self-sufficiency in metropolitan areas is commonly observed.

Thirdly, the results reveal the location-specific capacity of different agricultural land types in the PRD and provide valuable information on meeting local demands in food through the utilization of adjacent agricultural areas. The foodsheds of the regions with horticultural crops were the smallest and were located within the PRD, demonstrating a strong localized production capacity. Meanwhile, the foodshed of the arable lands, the main land type for food production, indicates the large spatial extension of the arable land areas toward the rural periphery, as the MCR requires a large area of additional arable land to meet its food consumption [9]. Moreover, we interviewed the managers of some wholesale markets and the Guangdong Logistics Association. According to them, about 60% to 70% of agricultural products come from areas adjacent to the PRD. This experience is significantly in line with the arable land foodshed analysis results. It needs to be mentioned that policymakers ought to consider that not all production areas within the foodshed can be used to feed the PRD, as these areas should first satisfy their own demands [12,26]. As multiple densely populated areas are present towards the east and north of the PRD, including the Chaozhou–Shantou area (Guangdong Province) and Ganzhou (Jiangxi Province), we should carefully consider the food demands of these areas when determining the major food production regions. To summarize, as Świąder et al. [25] and Zasada et al. [9] have pointed out, urban agriculture is not sufficient to feed all city residents, and urban consumption centers are increasingly dependent on agricultural production areas from the hinterlands surrounding cities, which is also applicable and in agreement with our observation in the PRD-MCR. In our study, by defining the foodshed as well as exploring the functional connections between the PRD and its hinterland, potential opportunities for supporting food production and localized food systems can be established in the PRD-MCR.

Our results differ from those of other studies when examining the size of foodsheds. Zasada et al. [9] applied the MFSS model to four European metropolitan areas of London, Berlin, Milan, and Rotterdam, finding that the foodshed in the London metropolitan area is only 91 km, despite the fact that London has the highest demand for agricultural land among the four cases. Similarly, Vicente-Vicente et al. [12] reported that the maximum radii of the foodsheds in Vienna and Bristol are 87.36 km and 37.34 km, respectively; however, in our study the foodshed with regard to the arable land reached 397.06 km. Compared to small cities (with populations of 1.79 million in Vienna and 449,300 in Bristol) or single-center metropolitan areas (with a population of slightly higher than 8 million in London), the foodshed areas estimated in our study are apparently much larger. This is because the MCR has a very high population (exceeding 78 million in the PRD) and multiple competing urban centers. The foodsheds of the various cities within the MCR are significantly overlapping, resulting in competition for the same agricultural resources, hence leading to a larger area of the adjacent hinterland being necessary to reach food sufficiency [7].

Finally, the results also revealed the uneven urban self-sufficiency levels within the MCR. Although overall self-sufficiency has been achieved in the PRD regarding certain agricultural land, there are significant differences between the cities within the PRD. More specifically, in the PRD the self-sufficiency regarding the production of horticultural crops from the respective regions exceeds 100%; however, cities such as Shenzhen, Foshan, Dongguan, Zhuhai, and Zhongshan are still not self-sufficient. At the same time, the self-sufficiency rate regarding different agricultural land types and their respective products varies greatly between cities. In general, central/big cities with a high population density, such as Shenzhen, Guangzhou, Dongguan, and Foshan, have lower self-sufficiency rates, while non-central/small cities, such as Zhaoqing, Jiangmen, Huizhou, and Zhuhai, have relatively higher self-sufficiency rates. In the PRD-MCR, the presence of multiple central cities with lower self-sufficiency rates greatly reduces the overall self-sufficiency level, posing more challenges to regional food system planning.

In recent years, the urbanization in the PRD-MCR has been accelerating, accompanied by a population surge. Urban areas have been continuously expanding, seriously squeezing out land used for agricultural production [38,48]. At the same time, due to the increased food consumption, changes in food preferences, and the increased diversification of residents’ diets, regional food consumption has begun to shift from being plant-based to being balanced plant/animal-based [49]. This has also led to an increased consumption of foods such as meat, poultry, eggs, and aquatic products [50]. These changes in food preferences and consumption have created a heightened demand for food supply, with higher demand for animal-based food, which in turn requires more agricultural land resources (especially arable land) [49]. In addition, considering that the PRD residents have a preference for fresh fruits, vegetables, and aquatic products in their dietary habits, there is consequently a significant increase in demand for areas for horticultural crop production and aquaculture in water bodies.

Certain initiatives can be undertaken to improve regional self-sufficiency to achieve a more sustainable and resilient food system. According to Li et al. [40], the agricultural land in the PRD is typically characterized by small, fragmented farms, fields, and non-flat fields. In recent years, some local governments have introduced a series of policies to reduce the fragmentation of arable land and have attempted to increase the amount of arable land and its quality. These policies include agricultural land consolidation, high-standard farmland construction as well as management, and basic farmland protection [51]. In general, facing the severe shortage of arable land in the MCR, an effective way to conserve these resources is to utilize the land in the hilly areas as much as possible (e.g., the use of land in hills to plant fruit trees) [40]. Optimizing crop rotation patterns and increasing crop diversification to improve the yield per unit of arable land area are other common countermeasures [52,53]. The application of alternative food networks, via the expansion of facility agriculture (solar greenhouses, plant factories), zero-acreage farming (such as roofs/indoor areas) [54], community gardens [55,56], community-supported agriculture (CSA) [23], and urban agriculture [57] is also acknowledged to have the potential to effectively increase food production output [9,58,59]. In addition, according to the current dietary patterns, the per capita consumption of red meat by local residents in the PRD exceeds the recommended standard of the Chinese Dietary Guidelines by approximately three-fold [60]. Therefore, the residents should be encouraged to adopt resource-saving dietary patterns, especially to reduce the consumption of animal products. These are conducive to the transformation of food production from different land resources instead of heavily depending on one land type (e.g., arable land) [23,53,61].

Although we have extensively explored foodsheds and self-sufficiency in the PRD, our study still has certain limitations. Firstly, our study did not consider food loss and waste due to the lack of relevant data. Research has shown that reducing food loss and waste can increase self-sufficiency by 20–30% [39,62,63,64], effectively alleviating the degree of ‘hunger’ in cities; however, difficulties in obtaining food loss data hinder the implementation of policies for reducing food losses in many countries [65]. The effective acquisition of data on food losses and waste from ‘farm to fork’ can be considered a critical future research objective, as it is fundamental for exploring and implementing novel food loss reduction technologies, policies, and planning methods. Secondly, the consumption data for different types of food come from the Guangdong Provincial Statistical Annals, and we have used the same food consumption standards for all of the residents of the area of study; however, food consumption may differ across cities and depend on the age group of the population [29]. Thus, further researching and identifying the dietary preferences of different population groups in different regions is another research direction that should be followed. Thirdly, a foodshed analysis serves as a potentially important tool for raising awareness and providing information for policy analysis [66]; however, data limitations, especially on local food consumption patterns, food flow, and the resolution of land cover data, are major hurdles that restrict foodshed analysis from qualitative to quantitative studies on urban food supply chains. Therefore, it would be beneficial to systematically investigate the underlying food transport flows between the production areas and urban consumption centers. As Schreiber et al. [20] stated, by drawing a common framework and coherent set of methodological criteria, future urban foodshed research can more readily contribute to informing policies to address the sustainability and resilience of food systems. In addition, Lăzăroiu et al. [67] indicated that organic food production and consumption offer health and sustainability upsides. Therefore, motivating farmers to engage in organic agriculture and seeking to establish an organic production system will benefit decisional aspects of sustainable food production.

## 5. Conclusions

Based on an extensive review of recent and relevant literature, we evaluated the self-sufficiency and foodshed capacity of different types of agricultural land in the PRD-MCR, which hold great significance for sustainable regional food systems and food security. In the practical case study on the PRD, a typical MCR, we emphasized the importance of distinguishing different types of agricultural land and provided a new understanding of the comprehensive characteristics of regional self-sufficiency. The results indicated significant differences in the self-sufficiency levels and foodshed sizes of the different agricultural land types in the study area. More specifically, self-sufficiency has been achieved at the regional level regarding horticultural crop production; however, no sufficiency has been achieved for water bodies, as their foodshed extends over the boundary of the PRD. Arable land exhibited the lowest self-sufficiency rate, resulting in a serious food shortage, both at the regional and urban levels. In terms of arable land, it is necessary to expand the scope of the foodshed to establish a self-sufficient local food system. In addition, the results of a foodshed assessment will help in the identification, with great precision, of areas with the highest agricultural production capacity and provide a basis with which to guide local planning, production, and consumption. An in-depth foodshed assessment can provide a useful experience (e.g., which types of food should interact with surrounding areas) for policymakers in establishing a local food system.

In this study, we indicate that, for MCRs, with a high population density and limited agricultural land, achieving food self-sufficiency faces increasingly complex challenges in the forthcoming ages. To conclude, our research findings can significantly contribute to the efficient planning of regional food systems and can guide the MCR’s local authorities to improve their policies toward the sustainable development of food systems.

## Figures and Tables

**Figure 1 foods-12-04210-f001:**
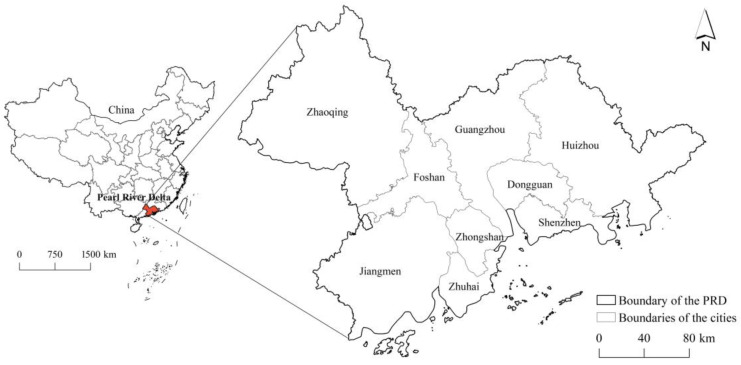
Study area—the Pearl River Delta megacity region.

**Figure 2 foods-12-04210-f002:**
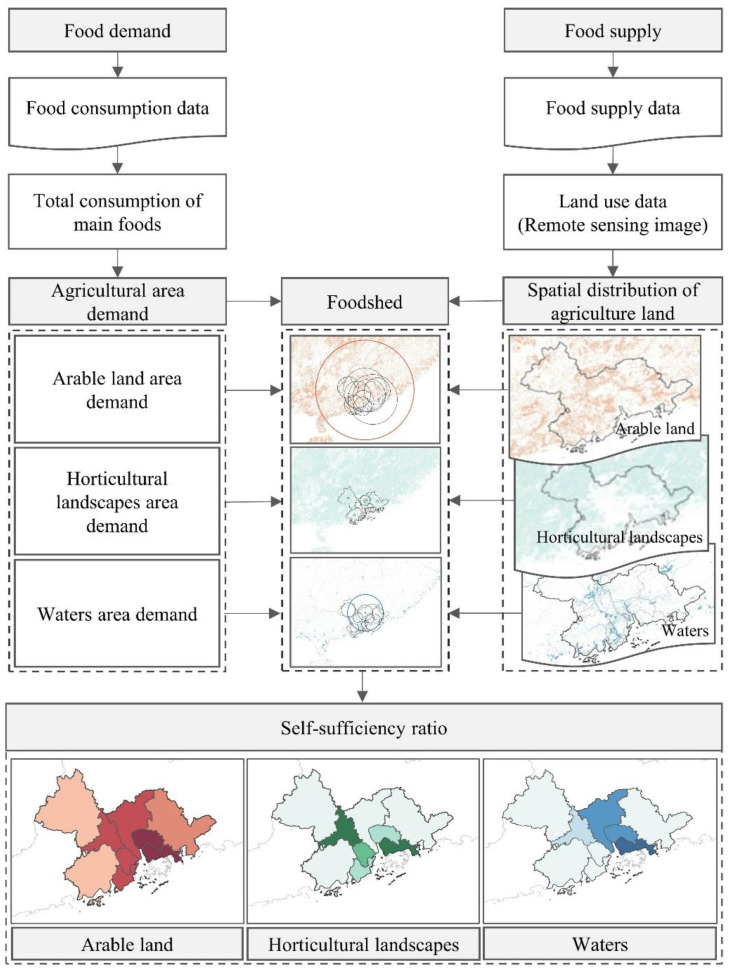
Methodological framework.

**Figure 3 foods-12-04210-f003:**
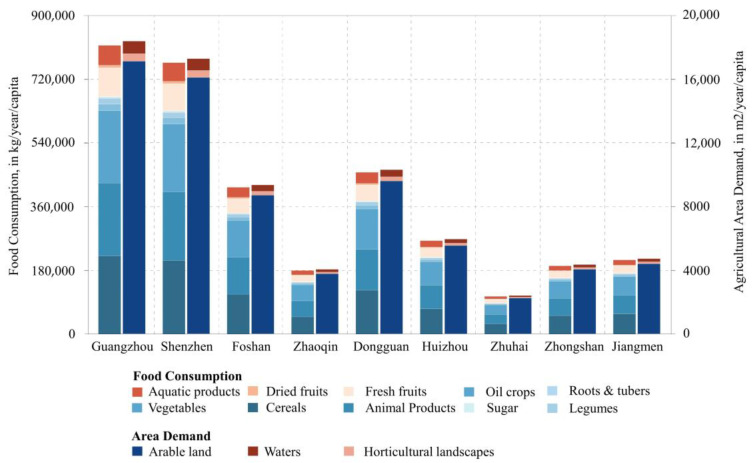
Food consumption and area demand for various agricultural land types in the PRD.

**Figure 4 foods-12-04210-f004:**
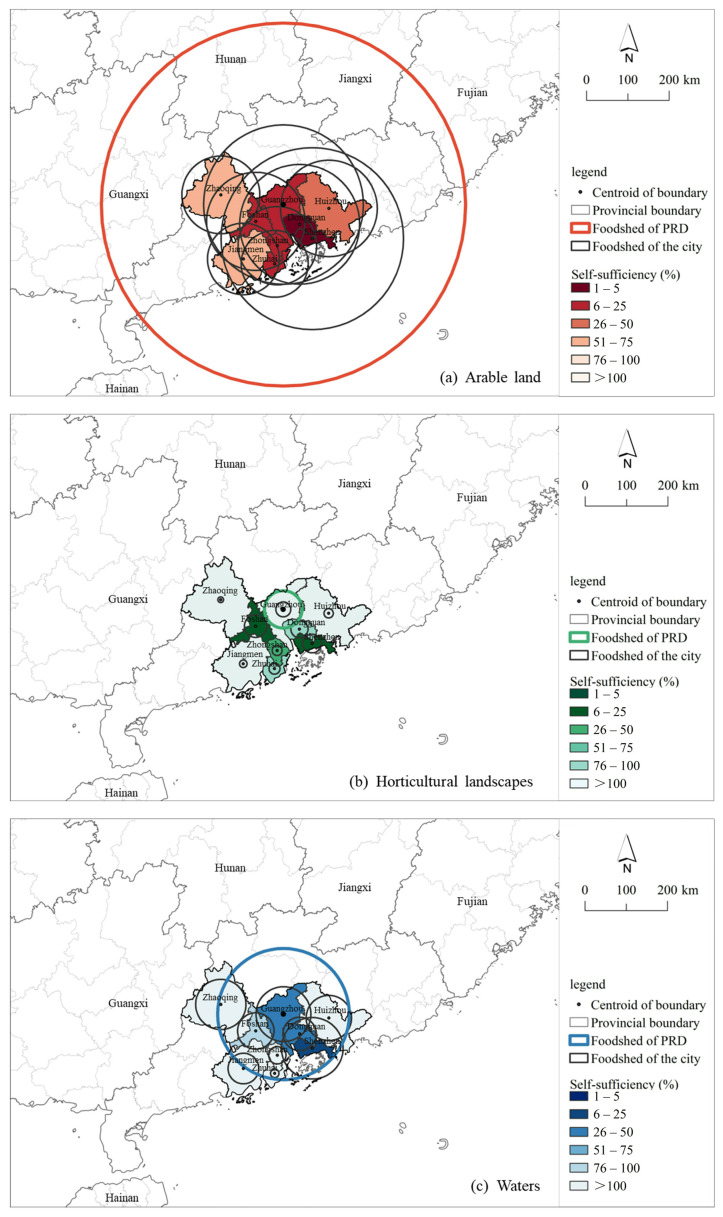
Foodshed and self-sufficiency level of the PRD. (**a**) shows the foodshed size and self-sufficiency of arable land in the PRD at the regional and urban levels respectively. Similarly, (**b**) and (**c**) respectively show the relevant information of the horticultural areas and waters.

**Table 1 foods-12-04210-t001:** Detailed information of the data.

Number	Name	Description	Source
1	Food consumption data (2021)	Per capita consumption of major foods in the Pearl River Delta (Unit: kg/year)	Guangdong Statistical Yearbook [41] (https://stats.gd.gov.cn/ (accessed on 22 July 2023))
2	Population data (2021)	The number of permanent residents in various cities in the Pearl River Delta	Chinese urban statistical yearbooks [42] (http://www.stats.gov.cn/ (accessed on 22 July 2023))
3	Land cover data (2020)	Land use in the Pearl River Delta (30 m resolution)	Data Center for Resources and Environmental Sciences, Chinese Academy of Sciences (RESDC) (http://www.resdc.cn/ (accessed on 22 July 2023))

**Table 2 foods-12-04210-t002:** Calculation of the conversion rates between food consumption and agricultural land demand.

Food Type		Conversion Factors of Crop–Agricultural Land	Feed Conversion Rate	Area of Agricultural Land Demanded to Produce 1 kg of Food	Type of Agricultural Land Demanded
Cereals		1.59	—	1.59	Arable land
Roots and tubers		2.08	—	2.08	Arable land
Legumes		5.07	—	5.07	Arable land
Oil crops		12.58	—	12.58	Arable land
Vegetables		0.28	—	0.28	Arable land
Pork		1.94	0.25 [44,45]	7.92	Arable land
Beef/mutton		2.63	0.32 [44,45]	8.35	Arable land
Poultry		2.29	0.35 [44,45]	6.54	Arable land
Aquatic products		2.46	2.22 [44,45]	1.23, 1.11	Waters/Arable land
Eggs		2.29	0.60 [44,45]	3.82	Arable land
Dairy		2.63	1.82 [44,45]	1.45	Arable land
Sugar		1.03	—	1.03	Arable land
Fruits	Fresh fruits	0.43	—	0.43	Horticultural landscapes
	Dried fruits	1.72	—	1.72	Horticultural landscapes

**Table 3 foods-12-04210-t003:** Total demand and self-sufficiency rate of various agricultural land types.

City	Population (Million)	Arable Land			Horticultural Landscapes			Waters		
		Self-Sufficiency Rate (%)	Demand (km^2^)	Radius (km)	Self-Sufficiency Rate (%)	Demand (km^2^)	Radius (km)	Self-Sufficiency Rate (%)	Demand (km^2^)	Radius (km)
Guangzhou	1867.66	11.47	17,119.30	174.24	35.64	475.67	16.32	162.96	789.28	59.85
Shenzhen	1756.01	0.75	16,095.85	198.23	5.46	447.24	20.45	22.99	742.10	67.27
Foshan	949.89	13.97	8706.82	107.42	85.22	241.93	24.53	13.69	401.43	39.82
Zhaoqing	411.36	60.80	3770.59	84.99	234.23	104.77	6.22	462.65	173.84	54.93
Dongguan	1046.66	2.70	9593.89	130.91	25.67	266.57	18.71	76.19	442.33	36.42
Huizhou	604.29	48.21	5538.98	105.58	121.35	153.90	10.17	548.69	255.37	49.52
Zhuhai	243.96	10.85	2236.17	73.21	295.88	62.13	12.05	90.33	103.10	9.23
Zhongshan	441.81	12.03	4049.67	85.71	102.39	112.52	10.85	47.30	186.71	18.95
Jiangmen	479.81	61.58	4398.01	78.33	386.11	122.20	8.22	262.31	202.77	33.51
PRD	7801.43	16.73	71,509.28	397.06	214.96	1986.94	40.97	84.13	3296.93	143.56

## Data Availability

The data used to support the findings of this study can be made available by the corresponding author upon request.

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
