# Peer review of "Assessing the Foodshed and Food Self-Sufficiency of the Pearl River Delta Megacity Region in China"

_foods, 2023, doi:10.3390/foods12234210_

Round 1

Reviewer 1 Report

Comments and Suggestions for Authors

Thank you for the opportunity to review the manuscript entitled “Assessing foodshed and food self-sufficiency of Pearl River Delta mega city region in China”.

The abstract should be rephrased, it appears rather redundant and the purpose of the research should be highlighted and clarified. In addition, which are the methods adopted in the research? Please, specify them starting from the abstract. 

L. 34. Considering that UN represents an acronym, please include (the first time you cite it in the text) the entire wording, namely “United Nations”. 

In the section “Materials and methods”, the authors should provide more systematic information related to the data collection (LL. 148-153).

L. 157. Could the authors provide more details related to the “food-land footprint”?

Could the authors please include the source for the different conversion rates provided in Table 1?

L. 191. Is there a reference for the “official statistical yearbook”?

“Results” are clearly presented, as well as “Discussion”. The manuscript is interesting and original. 

Comments on the Quality of English Language

English is clear and required a minor spell and grammar check. For instance, L. 13 presents a dot instead of a comma, whereas L. 14 misses a dot in concluding the sentence. 

Author Response

Many thanks for your valuable suggestions and comments, which does make the manuscript more readable and reasonable after revisions. We tried our best to improve the manuscript within the limited time and made some changes marked in red in revised paper which will not influence the content and framework of the paper. We appreciate for your warm work earnestly and hope the correction will meet with approval. Once again, thank you very much for your comments and suggestions. In the revised version, changes to our manuscript were all highlighted within the document by using red-colored text. The point-to-point response to your comments and suggestions has been added to the attachment.

Reviewer 2 Report

Comments and Suggestions for Authors

Replace it/they with the proper words to avoid confusion. E.g., ‘It can aid in determining the area of agricultural land that a city or region would theoretically need’.

The in-text citation system is variable, either with name + number (although they must be close to each other), or with number. E.g., ‘According to Vicente-Vicente et al., a foodshed can be considered as the territory around an urban/metropolitan area that is required to feed its population [18], indicating the connections between the urban/metropolitan area and its surrounding agricultural production hinterland [19,20].’

Also, ‘indicating’? By whom, by Vicente-Vicente et al.? If yes, why mentioning [19,20]? More development and depth of the methodology and analysis are needed.

The author should work harder on the approach adopted, establish a clear theoretical background to contextualize the analysis and narrow the scope of the analysis to specific aspects.

You claim: ‘In this context, Zasada et al. proposed the Metropolitan Foodshed and Self-sufficiency Scenario (MFSS) model with an assessment capacity at the city level [9]. Following their work, numerous studies extended the MFSS and similar tools to the regional level to support specific strategies for the management of the localized food system [12,22,25].’ [9] was published in 2019. [22] was published in 2018. [25] in 2019. How were they ‘following their work’ by publishing a year before, respectively in the same year?

'high population density and scarcity of agricultural land' – whose is this quote? The figures require more explanations.

‘Moreover, we also conducted interviews on some wholesale markets and the Guangdong Logistics Profession Association, which revealed that about 60% -70% of agricultural products come from areas adjacent to the PRD.’ – provide data and methodology.

‘as ÅšwiÄ…der et al. and Zasada et al. pointed out, urban agriculture is not sufficient to feed all city residents, and urban consumption centers are increasingly dependent on agricultural production areas from the hinterland surrounding the cities [9,22]’ – mention the numbers close to their sources, as here they are vice versa.

‘consumption of 'red meat' by local residents’ – why in quotes? There is a need of structuring the discussion to ensure that the methodological aspects are clearly presented.

The conclusion, too short, should clarify the main contribution of the paper and the value added to the field. Conclusion needs to be rewritten so that only important results are brought out along with their interpretation, comparison with earlier studies, and implications in a more integrated fashion.

Some of the cited sources are not properly edited.
The relationship between sustainable regional food systems and supply chain food loss as regards foodshed and food self-sufficiency has not been covered, and thus such sources can be cited:

Majerova, J.; Sroka, W.; Krizanova, A.; Gajanova, L.; Lazaroiu, G.; Nadanyiova, M. Sustainable Brand Management of Alimentary Goods. Sustainability 2020, 12, 556. https://doi.org/10.3390/su12020556

Pocol, C.B., Amuza, A., Moldovan, M.G., Stanca, L., Dabija, D.C. 2023. Clustering food wasters on an emerging market: a national wide representative research. Foods, 12(10), 1973. https://doi.org/10.3390/foods12101973

Lăzăroiu, G., Andronie, M., Uţă, C., and Hurloiu, I. (2019). “Trust Management in Organic Agriculture: Sustainable Consumption Behavior, Environmentally Conscious Purchase Intention, and Healthy Food Choices,” Frontiers in Public Health 7: 340. doi: 10.3389/fpubh.2019.00340.

Author Response

(The authors gave the same response as above.)

Round 2

Reviewer 2 Report

Comments and Suggestions for Authors

This revised version can be published.